# *Muricauda okinawensis* sp. Nov. and *Muricauda yonaguniensis* sp. Nov., Two Marine Bacteria Isolated from the Sediment Core near Hydrothermal Fields of Southern Okinawa Trough

**DOI:** 10.3390/microorganisms11061580

**Published:** 2023-06-14

**Authors:** Wenrui Cao, Xingyu Deng, Mingyu Jiang, Zhigang Zeng, Fengming Chang

**Affiliations:** 1Key Laboratory of Marine Geology and Environment, Institute of Oceanology, Chinese Academy of Sciences, Qingdao 266071, China; dengxingyu@qdio.ac.cn (X.D.); myjiang@qdio.ac.cn (M.J.); zgzeng@qdio.ac.cn (Z.Z.); chfm@qdio.ac.cn (F.C.); 2College of Earth Science and Engineering, Shandong University of Science and Technology, Qingdao 266590, China

**Keywords:** *Flavobacteriaceae*, *Muricauda okinawensis* sp. nov, *Muricauda yonaguniensis* sp. nov., polyphasic taxonomy, hydrothermal field

## Abstract

Two strains, 81s02^T^ and 334s03^T^, were isolated from the sediment core near the hydrothermal field of southern Okinawa Trough. The cells of both strains were observed to be rod-shaped, non-gliding, Gram-staining negative, yellow-pigmented, facultatively anaerobic, catalase and oxidase positive, and showing optimum growth at 30 °C and pH 7.5. The strains 81s02^T^ and 334s03^T^ were able to tolerate up to 10% and 9% (*w*/*v*) NaCl concentration, respectively. Based on phylogenomic analysis, the average nucleotide identity (ANI) and the digital DNA-DNA hybridization (dDDH) values between the two strains and the nearest phylogenetic neighbors of the genus *Muricauda* were in range of 78.0–86.3% and 21.5–33.9%, respectively. The strains 81s02^T^ and 334s03^T^ shared 98.1% 16S rRNA gene sequence similarity to each other but were identified as two distinct species based on 81.4–81.5% ANIb, 85.5–85.6% ANIm and 25.4% dDDH values calculated using whole genome sequences. The strains 81s02^T^ and 334s03^T^ shared the highest 16S rRNA gene sequence similarity to *M. lutimaris* SMK-108^T^ (98.7%) and *M. aurea* BC31-1-A7^T^ (98.8%), respectively. The major fatty acid of strains 81s02^T^ and 334s03^T^ were identified similarly as iso-C_15:0_, iso-C_17:0_ 3-OH and iso-C_15:1_ G, and the major polar lipids of the both strains consisted of phosphatidylethanolamine and two unidentified lipids. The strains contained MK-6 as their predominant menaquinone. The genomic G+C contents of strains 81s02^T^ and 334s03^T^ were determined to be 41.6 and 41.9 mol%, respectively. Based on the phylogenetic and phenotypic characteristics, both strains are considered to represent two novel species of the genus *Muricauda*, and the names *Muricauda okinawensis* sp. nov. and *Muricauda yonaguniensis* sp. nov. are proposed for strains 81s02^T^ (=KCTC 92889^T^ = MCCC 1K08502^T^) and 334s03^T^ (=KCTC 92890^T^ = MCCC 1K08503^T^).

## 1. Introduction

The genus *Muricauda*, a member of the family *Flavobacteriaceae* within the class *Flavobacteriia*, was first proposed by Bruns et al. [1], with *Muricauda ruestringensis* as the type species, and subsequently emended by Yoon et al. [2], Hwang et al. [3] and Liang et al. [4]. Consequently, *Flagellimonas* and *Spongiibacterium* were transferred to genus *Muricauda* [4,5], and *Flagellimonas algicola* [6], *Flagellimonas pacifica* [7,8], *Flagellimonas maritima* [9], *Flagellimonas aquimarina* [8], *Flagellimonas eckloniae* [10] and *Spongiibacterium flavum* [8,11] were reclassified as *Muricauda algicola* [4], *Muricauda parva* [4], *Muricauda aurantiaca* [4], *Muricauda koreensis* [5], *Muricauda eckloniae* [5] and *Muricauda flava* [5], respectively. On the other hand, one species of genus *Muricauda*, named *Muricauda lutea* [12], was reclassified as *Croceivirga lutea* [13]. At present, the genus *Muricauda* comprises 37 species with valid published and correct names and two species with not validly published names (https://lpsn.dsmz.de/genus/muricauda, accessed on 1 May 2023). Most of the these species were isolated from marine environments, such as seawater [3,14,15,16,17], tidal flats [18,19,20], coast sand [21], mangrove wetland [22], deep marine sediments [23,24,25], marine plants [6,8,10] and marine animals [11,26,27,28]. Members of genus *Muricauda* were characterized as Gram-stain negative, rod-shaped and having a DNA G+C content of 38–57 mol%, with MK-6 as the major isoprenoid quinone.

The deep-sea hydrothermal field represents one of the most physically and chemically diverse biomes on the Earth [29,30]. The Okinawa Trough (OT) contains several active hydrothermal fields, altogether making it an interesting region. A few strains had been isolated from the hydrothermal fields of the OT, where the composition of prokaryotic communities in different regions was also investigated [31,32,33,34,35,36,37,38]. It was suggested that *Muricauda* had never been the main group of microorganisms in the seafloor hydrothermal fields, although it did exist in the environment. In this study, during the investigation of the bacterial community from the sediment core near the hydrothermal fields of southern OT, two strains (designated 81s02^T^ and 334s03^T^), representing two novel members of the genus *Muricauda*, were characterized after isolation and purification via polyphasic taxonomy and comparative genome analysis.

## 2. Materials and Methods

### 2.1. Sampling, Isolation and Maintenance

The samples were recovered by a gravity corer from southern OT during the HOBAB4 cruise of the R/V Kexue at Station S2 in 2016. Sediment core HOBAB4-S2 (24°52′49.91″ N; 122°37′19.70″ E; water depth, 1505 m) was collected from a rifted basin between the Yonaguni Knoll IV and Tangyin hydrothermal field. Marine sediment dilutions (up to 10^−2^, 0.1 g sediment in 9.9 mL artificial sea water) were spread-plated on marine agar 2216 (MA, pH 7.2; BD Difco, New York, NY, USA) [39] and incubated at 25 °C under aerobic condition. Two isolates, designated as 81s02 and 334s03, were obtained after being incubated for 7 days from the samples of 81 cm and 334 cm below surface, respectively. The strains were stored at −80 °C in 20% (*v*/*v*) glycerol. The type strains *Muricauda ruestringensis* B1^T^ (=DSM 13258^T^), *Muricauda aurea* BC31-1-A7^T^ (=MCCC M23246^T^), *Muricauda aquimarina* SW-63^T^ (=JCM 11811^T^) and *Muricauda lutimaris* SMK-108^T^ (=KCTC 22173^T^), purchased from the Marine Culture Collection of China (MCCC), were used as reference strains for comparative purposes. Unless otherwise described, the strains were cultivated on marine agar 2216 (MA; Difco) at pH 7.5 and 30 °C.

### 2.2. 16S rRNA Gene Sequence and Phylogeny

Genomic DNA was extracted using a genomic DNA extraction kit (TIANGEN Biotech Co., Ltd., Beijing, China) according to the manufacturer’s instructions. The 16S rRNA gene was amplified by the universal primers 27F and 1492R, as previously described [40]. The PCR product was purified and ligated into the PMD19-T vector (TaKaRa, Kusatsu, Japan) and cloned according to the manufacturer’s instructions. Sequencing was performed by BGI (Qingdao, China). A BLAST search (https://www.ncbi.nlm.nih.gov, accessed on 10 April 2023) and the EzTaxon-e server (http://www.ezbiocloud.net, accessed on 10 April 2023) were used to calculate the pairwise sequence similarity based on the almost complete 16S rRNA gene sequences [41]. The multiple alignments of sequences for the new strains (81s02^T^ and 334s03^T^) and other type strains of the most closely related species were performed using CLUSTAL X [42]. A phylogenetic analysis was subsequently performed and the phylogenetic trees were constructed with MEGA version X [43] using the neighbor-joining (NJ) [44], maximum parsimony (MP) [45] and maximum likelihood (ML) [46] algorithms. The bootstrap analyses were based on 1000 re-samplings and the complete deletion option was used for the analysis [47], and the distances were calculated according to the two-parameter model of Kimura [48].

### 2.3. Genomic Characterization

The draft genomes of strains 81s02^T^ and 334s03^T^ were sequenced using Allwegene Tech Co. Ltd. (Beijing, China), using the Illumina HiSeq platform. A raw sequencing data assembly was performed using SOAPdenovo [49,50]. Open Reading Frames (ORFs) were predicted using prodigal (v2.6.3) [51] and the identification of tRNAs and rRNAs was carried out using tRNAscan-SE (v1.3.1) [52]. Protein coding regions were annotated against the Kyoto Encyclopedia of Genes and Genomes (KEGG) database [53]. The assignments of key metabolic pathways and specific functions were manually verified based on the KEGG result and the online KEGG mapping tools (https://www.genome.jp/kegg/kegg1b.html, accessed on 12 May 2023). The functional annotation of the genomes was also performed with rapid annotations using subsystems technology (RAST) to estimate genes involved in different categories (https://rast.nmpdr.org/, accessed on 12 May 2023) [54,55]. The Carbohydrate-Active Enzymes database (http://www.cazy.org/, accessed on 13 May 2023) [56] was used to predict carbohydrate-active enzymes. To classify the position of 81s02^T^ and 334s03^T^ in the genus *Muricauda* using genome sequence-based comparison, the digital DNA–DNA hybridization (dDDH) value was calculated using the Genome-to-Genome Distance Calculator 3.0 (http://ggdc.dsmz.de/ggdc.php/, accessed on 26 April 2023) with the recommended parameter, Formula 2 [57]. Average nucleotide identity calculation based on Blast+ (ANIb) and MUMmer (ANIm) was estimated using JspeciesWS (http://jspecies.ribohost.com/jspeciesws/, accessed on 24 May 2023) [58]. The draft genome sequences of strain *M. abyssi* W52^T^ (JAHZSV000000000), *M. aquimarina* SW-63^T^ (RZMZ00000000), *M. aurea* BC31-1-A7^T^ (JAFLNL000000000), *M. brasiliensis* K001^T^ (QBTW00000000), *M. chongwuensis* HICW^T^ (WYET00000000), *M. lutimaris* SMK-108^T^ (QXFH00000000), *M. oceanensis* 40DY170^T^ (RZNA00000000), *M. oceani* 501str8^T^ (CP049616) and *M. ruestringensis* B1^T^ (CP002999) were obtained from the GenBank database for the overall genome-related index (OGRI) analyses [59]. The draft genomes of two *Muricauda* strains in this study and close relatives affiliated with the genus *Muricauda* were downloaded from the NCBI GenBank database. The 120 conserved concatenated proteins (Bac120 sets) were identified using GTDB-Tk v. 1.3.0 and used to construct a phylogenetic tree using FastTree [60,61,62]. *Formosa maritima* was used as an outgroup.

### 2.4. Morphological, Physiological and Biochemical Characterization

Cell morphology was observed using the 3-day cultures via light microscopy (BX53; Olympus, Shinjuku, Japan) and transmission electron microscopy (Hitachi-HT7700, Tokyo, Japan). Gram staining was determined using a Gram Stain kit (QingDao Hopebio-Technology Co., Ltd., Qingdao, China) according to the manufacturer’s instructions. Gliding motility was performed as previously described [63]. Oxidase activity was tested with oxidase reagent (bioMérieux, Marcy-l’Étoile, France) and catalase activity was examined by bubble production in 3% (*v*/*v*) hydrogen peroxide. Nitrate reduction and the hydrolysis of agar, casein, DNA, starch and Tween 80 were carried out according to a previous description [64]. The temperature range for growth was assessed by incubating the strains at 4–40 °C (4, 10, 16, 20, 25, 28, 30, 33, 35, 37 and 40 °C). Growth at different pH values (5.5, 6.0, 6.5, 7.0, 7.5, 8.0, 8.5, 9.0, 9.5 and 10) was performed in marine ZoBell broth (MB). For pH tolerance experiments, the following buffer solutions were used: MES (pH 5.5–6.0), PIPES (pH 6.5–7.0), HEPES (pH 7.5–8.0), Tricine (pH 8.5) and CAPSO (pH 9.0, 9.5 and 10.0) at a concentration of 20 mM. Growth at various NaCl concentrations was investigated in a modified MB made with 0.5% peptone, 0.1% yeast extract, 0.01% FePO_4_, artificial seawater (0.6% MgCl_2_, 0.32% Na_2_SO_4_, 0.18 % CaCl_2_, 0.06% KCl, 0.02% Na_2_CO_3_, traces of Na_2_SiO_3,_ and NaF, *w*/*v*) and in the presence of 0–10% (*w*/*v*) NaCl (0, 0.5, 1, 2, 3, 4, 5, 6, 7, 8, 9 and 10 %). Anaerobic growth (10% CO_2_, 5% H_2_ and 85% N_2_) and micro-aerobic growth (5% O_2_, 10% CO_2_ and 85% N_2_) were detected in an anaerobic chamber on MA plates with or without 0.25% (*w*/*v*) NaNO_3_ for 15 days at 30 °C. The antibiotic sensitivity of the strain was examined as recommended [65], using the antibiotic-impregnated discs (Hangzhou Microbial Reagent Co., Ltd., Hangzhou, China) in the following: norfloxacin (10 μg), gentamicin (10 μg), kanamycin (30 μg), cephalexin (30 μg), medilamycin (30 μg), chloramphenicol (30 μg), erythromycin (15 μg), clindamycin (2 μg), ampicillin (10 μg), penicillin (10 μg), ofloxacin (5 μg), vancomycin (30 μg), neomycin (30 μg), polymyxin B (300 IU), tetracycline (30 μg) and compound sulfamethoxazole (23.75/1.25 μg). Additional physiological and biochemical characteristics were tested using API 20NE, API 50CHB and API ZYM strips (bioMérieux), according to the manufacturer’s instructions, with the single modification of adjusting the salinity to 3% (*w*/*v*).

### 2.5. Chemotaxonomic Characterization

The cell biomass of strains for chemotaxonomic analysis was collected and freeze-dried after incubation in MB at 30 °C for 48 h. The respiratory isoprenoid quinones of strains 81s02^T^ and 334s03^T^ were extracted and analyzed by HPLC as described [66,67,68]. The polar lipids of novel isolates were extracted and detected using 2D thin-layer chromatography (TLC). Total lipid material was examined using molybdatophosphoric acid and specific functional groups were investigated using spray reagents specific for each, according to Tindall et al. [69]. The fatty acids were determined for the strains 81s02^T^ and 334s03^T^, as well as for *M. aquimarina* SW-63^T^, *M. aurea* BC31-1-A7^T^, *M. lutimaris* SMK-108^T^ and *M. ruestringensis* B1^T^, according to the standard protocol of the Microbial Identification System (MIDI, Sherlock Version 6.3). Fatty acids were methylated and analyzed using an Agilent 6890 N gas chromatography instrument (Santa Clara, CA, USA) and identified using the RTSBA6 database of the microbial identification system [70].

## 3. Results and Discussion

### 3.1. Phylogenetic and Genome Analysis

The almost full-length 16S rRNA gene sequences of strains 81s02^T^ (1488 bp, OQ547168) and 334s03^T^ (1488 bp, OQ547169) were determined and confirmed with the draft genome sequence. The strains shared 98.1% 16S rRNA gene sequence similarity to each other. Sequence similarity values calculated for the strains 81s02^T^ and 334s03^T^ indicated the greatest degree of similarity to *M. lutimaris* SMK-108^T^ (98.7%) and *M. aurea* BC31-1-A7^T^ (98.8%), respectively. Moreover, strain 81s02^T^ exhibited 16S rRNA gene sequence similarities of 98.7, 98.6, 98.3, 98.2, 98.1, 98.1, 98.1 and 98.0% to the related strains *M. aquimarina* SW-63^T^, *M. ruestringensis* B1^T^, *M. aurea* BC31-1-A7^T^, *M. abyssi* W52^T^, *M. brasiliensis* K001^T^, *M. oceanensis* 40DY170^T^, *M. oceani* 501str8^T^ and *M. chongwuensis* HICW^T^, respectively; and strain 334s03^T^ exhibited 16S rRNA gene sequence similarities of 98.6, 98.5, 98.5, 98.5, 98.4, 98.4, 98.3 and 98.2% to the related strains *M. abyssi* W52^T^, *M. aquimarina* SW-63^T^, *M. ruestringensis* B1^T^, *M. brasiliensis* K001^T^, *M. chongwuensis* HICW^T^, *M. oceani* 501str8^T^, *M. lutimaris* SMK-108^T^ and *M. oceanensis* 40DY170^T^, respectively. In addition, strains 81s02^T^ and 334s03^T^ showed less than 98.0% 16S rRNA gene sequence similarity to the other representative members within the genus *Muricauda*.

The draft genome sequence of strain 81s02^T^ (JARFVA000000000) resulted in 14 contigs and yielded a genome of 4,030,812 bp in length after assembly. Contigs varied in length from 1908 to 1,216,753 bp, and the N50 value was 852,588. The draft genome sequence of strain 334s03^T^ (JARFVB000000000) resulted in 44 contigs and yielded a genome of 4,292,830 bp in length after assembly. Contigs varied in length from 1164 to 968,529 bp, and the N50 value was 323,675. The sequencing depths of coverage were 347× and 294× for strains 81s02^T^ and 334s03^T^, respectively. The genomic DNA G+C contents of strains 81s02^T^ and 334s03^T^ were 41.6 and 41.9 mol%, respectively, which were determined from the genome sequence. ANIb and ANIm values between new strains (81s02^T^ and 334s03^T^) and reference strains ranged from 78.0% to 86.3%, which were significantly lower than the threshold value (95–96%) for the delineation of genomic species [71]. The dDDH values based on the draft genomes between new strains and reference strains ranged from 21.5% to 33.9%, which were far below cut-off values (70%) for species differentiation [57] (Appendix A). Moreover, the ANIb, ANIm and dDDH values between strain 81s02^T^ and 334s03^T^ were 81.4–81.5, 85.5–85.6 and 25.4%, respectively. The overall topological structures of the phylogenetic and phylogenomic trees (Figure 1 and Figure 2) clearly showed that strains 81s02^T^ and 334s03^T^ fell within the clade comprising species of the genus *Muricauda*. The phylogenetic trees using MP and ML algorithms also showed essentially the similar topology (Appendix A).

### 3.2. Morphological, Physiological and Biochemical Characteristics

Cells of the strains 81s02^T^ (0.4–0.6 × 1.0–3.0 μm) and 334s03^T^ (0.3–0.6 × 0.8–2.5 μm) were rod shaped after 3 days’ growth in MB (Figure 3). The two novel strains, appeared to be sensitive to cephalexin (30 μg), medilamycin (30 μg), clindamycin (2 μg) and ofloxacin (5 μg), and thus showed a similar antibiotic sensitivity. However, strain 334s03^T^ was sensitive to erythromycin (15 μg) and vancomycin (30 μg), while strain 81s02^T^ was not. Growth was observed in microaerobic conditions for both strains, but no growth was observed under anaerobic conditions. Strains 81s02^T^ and 334s03^T^ also shared several common biochemical properties with the reference strains *M. aquimarina* SW-63^T^, *M. aurea* BC31-1-A7^T^, *M. lutimaris* SMK-108^T^ and *M. ruestringensis* B1^T^. All abovementioned strains in this study were positive for the following: oxidase and catalase; the hydrolysis of aesculin and PNPG; acid production from d-glucose, d-fructose, d-mannose, MDM (methyl-*α*-d-mannopyranoside), MDG (methyl-*α*-d-glucopyranoside), amygdalin, arbutin, salicin, d-cellobiose, d-maltose, d-lactose, d-melibiose, sucrose, d-trehalose, d-melezitose, d-raffinose, starch, gentiobiose, d-turanose and potassium 5-ketogluconate (weak); and the activity of alkaline phosphatase, esterase lipase C4, esterase lipase C8, lipase (C14) (weak), leucine arylamidase, valine arylamidase, cystine arylamidase, acid phosphatase, naphthol-AS-BI-phosphohydrolase, *α*-galactosidase, *β*-galactosidase, *α*-glucosidase, *β*-glucosidase and *α*-mannosidase. All abovementioned strains in this study were negative for the following: Gram-reaction; flexirubin-type pigment test; hydrolysis of agar and DNA; acid production from glycerol, d-ribose, l-xylose, d-adonitol, d-mannitol and d-sorbitol; and activity of *α*-fucosidase. However, the two novel strains differed markedly from strains *M. aquimarina* SW-63^T^, *M. aurea* BC31-1-A7^T^, *M. lutimaris* SMK-108^T^ and *M. ruestringensis* B1^T^ having a positive result for the growth at pH 9.5 and acid production from glycogen. In addition, strain 81s02^T^ and 334s03^T^ could be distinguished from each other by the tolerance of NaCl, acid production from *N*-acetylglucosamine, activity of *N*-acetyl-glucosaminidase and their ability to hydrolyze casein. The complete morphological, physiological and biochemical characteristics of strains 81s02^T^ and 334s03^T^ were shown in the species description and Table 1.

### 3.3. Chemotaxonomic Characteristics

For both strains, predominant lipoquinone was menaquinone-6 (MK-6), which was in accordance with the previous observations for species of the genus *Muricauda* [1,2,3,16,17,18,25,28,72,73,74]. The major fatty acids (>10%) of strains 81s02^T^ and 334s03^T^ were iso-C_15:0_, iso-C_17:0_ 3-OH and iso-C_15:1_ G, followed by iso-C_15:0_ 3-OH. The fatty acid compositions of both strains was similar to those of closely related type strains within *Muricauda* (Table 2). Nevertheless, there were several different characteristics between these strains. For instance, the contents of iso-C_17:0_ 3-OH in 81s02^T^ and 334s03^T^ were significantly higher than those in *M. aquimarina* SW-63^T^, *M. aurea* BC31-1-A7^T^, *M. lutimaris* SMK-108^T^ and *M. ruestringensis* B1^T^. However, C_12:0_, C_15:0_ 3-OH and C_17:0_ 3-OH were absent in strains 81s02^T^ and 334s03^T^, while they were detected in the other type strains. The polar lipids of strain 81s02^T^ comprised phosphatidylethanolamine (PE), one unidentified aminolipid (AL) and two unidentified lipids (L 1–2). Strain 334s03^T^ comprised similar polar lipid components with strain J15B81-2^T^ except for the absence of one unidentified aminolipid (AL) (Appendix A). Phosphatidylethanolamine, phospholipid and two unidentified lipids (L 1–2) were the common major polar lipids of novel isolates and their related type strains within the genus *Muricauda*. Meanwhile, the number of ALs identified in strains 81s02^T^ and 334s03^T^ was less than that in *M. aquimarina* SW-63^T^, *M. lutimaris* SMK-108^T^ and *M. ruestringensis* B1^T^ [24].

**Table 1 microorganisms-11-01580-t001:** Differentiating characteristics of strains 81s02^T^ and 334s03^T^ from their closest phylogenetic relatives. Strains: (1) 81s02^T^; (2) 334s03^T^; (3) *M. lutimaris* SMK-108^T^; (4) *M. aurea* BC31-1-A7^T^; (5) *M. aquimarina* SW-63^T^; (6) *M. ruestringensis* B1^T^; (7) *M. brasiliensis* K001^T^ [73]; (8) *M. chongwuensis* HICW^T^ [72]; (9) *M. oceanensis* 40DY170^T^ [75]; (10) *M. oceani* 501str8^T^ [24]. Data for strains 1, 2, 3, 4, 5 and 6 were determined from this study unless indicated. +, positive; w, weakly positive; −, negative; ND, no data available.

Characteristic	1	2	3	4	5	6	7 ^♯^	8 ^♯^	9 ^♯^	10 ^♯^
Growth at:										
Temperature Range (°C)	10–40	10–40	10–38 *	10–40 *	10–44 *	8–40 *	15–37	15–40	15–40	8–42
Optimum temperature	30	30–35	30 *	28–32 *	30–37 *	20–30 *	28–33	25–30	35	25–30
pH Range	6.0–9.5	6.0–9.5	5.0–8.0 *	6.0–8.5 *	6.0–9.0 *	6.0–9.0 *	6.0–8.0	6.0–8.0	5.5–9.0	5.5–9.0
Optimum pH	7.0–7.5	7.5	7.0–8.0 *	7.5 *	7.0 *	8.0 *	ND	7	6.5–7.0	7
NaCl Range (%, *w*/*v*)	0.5–10.0	0.5–9.0	1.0–10.0 *	2.0–10.0 *	0.5–9.0 *	0.5–9.0 *	0.5–9.0	0.5–8.0	0.5–10.0	0.5–10.0
Nitrate reduction	−	−	−	+	w	−	−	w	−	−
Indole production	−	−	−	+	−	−	−	ND	−	−
Hydrolysis of:										
Arginine	−	−	−	+	−	−	ND	ND	−	+
Casein	+	−	+	−	+	+	ND	ND	ND	−
Gelatin	−	−	−	+	−	−	−	−	−	−
Starch	−	−	−	−	−	−	−	ND	w	−
Tween 80	+	+	−	−	+	+	+	ND	+	−
Urease	−	−	−	+	−	−	−	ND	−	+
Acid production from:										
l-Arabinose	−	w	w	+	−	w	ND	ND	+	ND
d-Xylose	−	w	w	+	w	+	ND	ND	+	ND
d-Galactose	+	w	w	+	w	w	ND	ND	+	ND
l-Rhamnose	−	−	−	+	−	−	ND	ND	w	ND
*N*-Acetyl-glucosamine	+	−	w	+	+	w	ND	ND	ND	ND
Inulin	−	−	−	−	+	w	ND	ND	ND	ND
Glycogen	+	+	w	−	−	−	ND	ND	ND	ND
Enzymic activities:										
Trypsin	+	+	−	+	+	+	+	+	+	+
α-Chymotrypsin	−	−	w	+	+	−	+	+	+	−
β-Glucuronidase	−	−	+	−	−	−	−	w	+	−
*N*-Acetyl-glucosaminidase	+	−	+	+	+	+	+	+	+	+
DNA G+C content (mol%)	41.6	41.9	40.1	42.1	43.4	41.4	41.6	41.4	42.4	42.8

* Data taken from the literature [1,2,18,74]. ^♯^ Data taken from the literature [24,72,73,75].

**Table 2 microorganisms-11-01580-t002:** Fatty acid content of strains 81s02^T^ and 334s03^T^ in comparison with those of closely related strains. Strains: (1) 81s02^T^; (2) 334s03^T^; (3) *M. aquimarina* SW-63^T^; (4) *M. aurea* BC31-1-A7^T^; (5) *M. lutimaris* SMK-108^T^; (6) *M. ruestringensis* B1^T^. All data were obtained in this study. Fatty acids amounting to above 10% were in bold. Fatty acids amounting to less than 0.50% of the total in all strains were not included. TR, trace (<0.50 %); ND, not detected.

Fatty Acid	1	2	3	4	5	6
Straight-chain:						
C_12:0_	ND	ND	TR	TR	0.56	TR
C16:0	1.95	2.07	0.74	0.93	0.54	1.25
C18:0	0.74	0.64	TR	ND	TR	TR
Branched:						
iso-C13:0	TR	TR	0.79	0.84	1.07	0.81
iso-C14:0	TR	ND	2.23	ND	4.76	5.00
iso-C15:0	**39.08**	**39.60**	**38.29**	**46.61**	**36.85**	**38.33**
iso-C15:1 G	**13.07**	**15.04**	**19.76**	**10.64**	**15.23**	**15.18**
iso-C16:0	0.77	0.53	2.43	1.35	0.68	1.00
iso-C17:0	0.77	TR	TR	1.00	TR	0.74
anteiso-C15:0	0.87	1.97	4.16	1.37	3.66	3.55
Unsaturated:						
C_18:1_ *ω*9*c*	0.54	0.70	TR	TR	TR	TR
Hydroxy:						
C15:0 3-OH	ND	ND	0.72	TR	1.77	0.59
C16:0 3-OH	0.68	0.61	TR	TR	0.58	0.64
C17:0 2-OH	TR	ND	0.83	TR	1.34	0.98
C17:0 3-OH	ND	ND	TR	TR	1.45	0.59
iso-C15:0 3-OH	5.16	5.05	4.21	4.74	5.53	4.49
iso-C16:0 3-OH	1.53	0.85	3.98	1.51	2.53	1.92
iso-C17:0 3-OH	**26.68**	**27.72**	**15.95**	**22.48**	**19.21**	**18.97**
Summed features: *						
2	0.59	0.59	TR	TR	TR	TR
3	2.05	2.00	1.28	1.72	0.96	1.34
9	1.93	0.60	0.66	1.79	TR	0.82

***** Summed features are fatty acids that cannot be resolved reliably from another fatty acid using the chromatographic conditions chosen. The MIDI system groups these fatty acids together as one feature with a single percentage of the total. Summed feature 2 contained C_14:0_ 3-OH and/or iso-C_16:1_ I; Summed feature 3 contained C_16:1_
*ω*7*c* and/or C_16:1_ *ω*6*c*; Summed feature 9 contained C_16:0_ 10-methyl and/or iso-C_17:1_
*ω*9*c*.

### 3.4. Genome Attributes and Comparative Genome Analysis

The draft genome of strain 81s02^T^ contained 3662 ORFs and 42 tRNAs, while strain 334s03^T^ contained 3864 ORFs and 38 tRNAs. Pathway analyses on the KEGG website suggested that all eleven strains in this study within genus *Muricauda* had a complete glycolysis pathway (Embden–Meyerhof pathway, EMP) although the genes encoding some reaction steps were diverse (Figure 4). A complete pentose phosphate pathway (PPP) was also found in all eleven genomes, and it was more conserved compared to the EMP, where all genes encoding the PPP found in all strains were the same, aside from *ripA*, which was only found in strain *M. oceani* 501str8^T^. On the other hand, all strains were lacking the gene *edd*, encoding phosphogluconate dehydratase, indicating that the Entner–Doudoroff pathway might not present in *Muricauda* strains. Several *Muricauda* species contained the genes *nasC* and *nirA*, such as *M. aurea*, *M. chongwuensis*, *M. brasiliensi*, *M. oceani* and *M. ruestringensis*, while the other species only contained *nirA* but not *nasC*. Commonly, genes *nasC* and *nirA* were recognized to reduce nitrate to nitrite and reduce nitrite to ammonia, respectively [76]. Therefore, *Muricauda* species had a diverse nitrate reduction capability. Moreover, a complete assimilatory sulfate reduction pathway was found in *M. ruestringensis* B1^T^ through the existence of genes *cysC*, *cysD*, *cysN*, *cysJ* and *cysI*. It was indicated that *M. ruestringensis* (the type species of genus *Muricauda*) had the capability of reducing sulfate to sulfite or hydrogen sulfide. However, other members in this study were lacking certain genes, showing an uncomplete sulfate reduction pathway. In addition, KEGG annotation showed that rhamnose containing glycans biosynthesis protein, polysaccharide biosynthesis/export protein, and lipopolysaccharide (dTDP-l-rhamnose) biosynthesis/assembly protein-related genes were found in all eleven *Muricauda* strains, and it has been suggested that it helps *Muricauda* survive in marine environments and assists them to endure extremes of temperature, salinity and nutrient availability. Metabolic features related to functional categories from RAST showed that genes associated with “virulence, disease and defense” existed in the genomes of *Muricauda* strains, which might be important for *Muricauda* to resist the toxic compounds in the environments (Appendix A). A large number of genes were involved in the class of “stress response”, which might provide these species the ability to adapt to special environments stresses, such as pressure, oxygen concentration, temperature, pH and salinity in marine ecosystem. 

Carbohydrate-active enzymes (CAZymes) are involved in many metabolic pathways and in the biosynthesis and degradation of various biomolecules, such as bacterial exopolysaccharides, starch, cellulose and lignin [77]. Thus, genes putatively coding for carbon metabolism were analyzed among different species within *Muricauda*, including CAZymes, redox enzymes with auxiliary activities (AAs) and those with carbohydrate-binding modules (CBMs) (Appendix A). CAZyme families were classified into four major groups: glycoside hydrolases (GHs), glycosyltransferases (GTs), polysaccharide lyases (PLs) and carbohydrate esterases (CEs). Strains 81s02^T^ and 334s03^T^ harbored similar CAZyme-encoding genes compared to other species of *Muricauda*. These genes were mainly present in the groups of GHs and GTs. A number of GH3, GH13, GH109, GT2 and GT4 were identified in all genomes; however, GH43 was not observed in the genome of strain 81s02^T^, which was present in other ten strains. Moreover, PL7, PL9 and CBM62 were only observed in the genome of strain *M. aurea* BC31-1-A7^T^, while PL8 was only observed in the genome of strain *M. chongwuensis* HICW^T^, suggesting a different process for carbon metabolism in these strains. Therefore, a more expanded investigation for the carbon utilization of the genus *Muricauda* is required to gain insights into a complete process for the carbon metabolism in the future.

## 4. Conclusions

Taken together, the phylogenetic analysis, chemotaxonomic data and phenotypic results presented above suggested that 81s02^T^ and 334s03^T^ represent two novel species of the genus *Muricauda*, for which the names *Muricauda okinawensis* sp. nov. and *Muricauda yonaguniensis* sp. nov. are proposed, with 81s02^T^ and 334s03^T^ as type strains, respectively.

### 4.1. Description of Muricauda okinawensis *sp. Nov.*

*Muricauda okinawensis* (o.ki.na.wen’sis. N.L. masc./fem. adj. okinawensis, pertaining to Okinawa Trough, where the type strain was isolated.) 

Cells are Gram-staining negative, facultatively anaerobic rods (0.4–0.6 × 1.0–3.0 μm) non-flagellated and non-gliding. Colonies on MA are circular, smooth, yellow-pigmented and about 1.5 mm in diameter after 2 days of growth at 30 °C. Growth occurs at 10–40 °C and in the presence of 0.5–10.0% (*w*/*v*) NaCl, with the optimum growth at 30 °C and with 2% (*w*/*v*) NaCl. The optimum pH is 7.0–7.5 and no growth occurs below pH 6.0 or above pH 9.5. A flexirubin-type pigment is not produced. It is positive for catalase and oxidase. Negative for nitrate reduction, indole and H_2_S production. It is positive for the hydrolysis of aesculin, casein, PNPG and Tween 80, but negative for the hydrolysis of agar, arginine, DNA, gelatin, starch and urea. Acid is produced from d-glucose, d-fructose, d-mannose, MDM, MDG, amygdalin, arbutin, salicin, d-cellobiose, d-maltose, d-lactose, d-melibiose, sucrose, d-trehalose, d-melezitose, d-raffinose, starch, gentiobiose, d-turanose, potassium 5-ketogluconate (weak), d-galactose, *N*-acetyl-glucosamine and glycogen. Alkaline phosphatase, esterase lipase C4, esterase lipase C8, lipase (C 14) (weak), leucine arylamidase, valine arylamidase, cystine arylamidase, trypsin, acid phosphatase, naphthol-AS-BI-phosphohydrolase, *α*-galactosidase, *β*-galactosidase, *α*-glucosidase, β-glucosidase, *N*-acetyl-glucosaminidase and *α*-mannosidase are present, while *α*-chymotrypsin, *β*-glucuronidase and *α*-fucosidase are absent. The major fatty acids are iso-C_15:0_, iso-C_17:0_ 3-OH and iso-C_15:1_ G. Menaquinone-6 (MK-6) is the predominant respiratory quinone. The major polar lipids are phosphatidylethanolamine and two unidentified lipids. The DNA G+C content of the type strain is 41.6 mol%.

The type strain is 81s02^T^ (=KCTC 92889^T^ = MCCC 1K08502^T^) and was isolated from the sediment core near the hydrothermal fields of the southern Okinawa Trough. The GenBank accession numbers for the 16S rRNA gene and the draft whole genome data of the strain 81s02^T^ are OQ547168 and JARFVA000000000, respectively.

### 4.2. Description of Muricauda yonaguniensis *sp. Nov.*

*Muricauda yonaguniensis* (Yo.na.gu.ni.en ‘sis N.L. masc./fem. adj. Yonaguniensis, pertaining to the Yonaguni Knoll IV hydrothermal field, where the type strain was isolated.) 

Cells are Gram-staining negative, facultatively anaerobic rods (0.3–0.6 × 0.8–2.5 μm), and are non-flagellated and non-gliding. Colonies on MA are circular, smooth, yellow-pigmented and about 1.5 mm in diameter after 2 days of growth at 30 °C. Growth occurs at 10–40 °C and in the presence of 0.5–9.0% (*w*/*v*) NaCl, with the optimum growth at 30–35 °C and with 2% (*w*/*v*) NaCl. The optimum pH is 7.5 and no growth occurs below pH 6.0 or above pH 9.5. A flexirubin-type pigment is not produced. It is positive for catalase and oxidase. It is negative for nitrate reduction, indole and H_2_S production. It is positive for the hydrolysis of aesculin, PNPG and Tween 80, but negative for the hydrolysis of agar, arginine, casein, DNA, gelatin, starch and urea. Acid is produced from d-glucose, d-fructose, d-mannose, MDM, MDG, amygdalin, arbutin, salicin, d-cellobiose, d-maltose, d-lactose, d-melibiose, sucrose, d-trehalose, d-melezitose, d-raffinose, starch, gentiobiose, d-turanose, potassium 5-ketogluconate (weak), d-galactose, l-arabinose (weak), d-xylose (weak) and glycogen. Alkaline phosphatase, esterase lipase C4, esterase lipase C8, lipase (C 14) (weak), leucine arylamidase, valine arylamidase, cystine arylamidase, trypsin, acid phosphatase, naphthol-AS-BI-phosphohydrolase, *α*-galactosidase, *β*-galactosidase, *α*-glucosidase, *β*-glucosidase and *α*-mannosidase are present, while *α*-chymotrypsin, *β*-glucuronidase, *N*-acetyl-glucosaminidase and *α*-fucosidase are absent. The major fatty acids are iso-C_15:0_, iso-C_17:0_ 3-OH and iso-C_15:1_ G. Menaquinone-6 (MK-6) is the predominant respiratory quinone. The major polar lipids are phosphatidylethanolamine and two unidentified lipids. The DNA G+C content of the type strain is 41.9 mol%.

The type strain is 334s03^T^ (=KCTC 92890^T^ = MCCC 1K08503^T^) and was isolated from the sediment core near the hydrothermal fields of the southern Okinawa Trough. The GenBank accession numbers for the 16S rRNA gene and the draft whole genome data of strain 334s03^T^ are OQ547169 and JARFVB000000000, respectively.

## Figures and Tables

**Figure 1 microorganisms-11-01580-f001:**
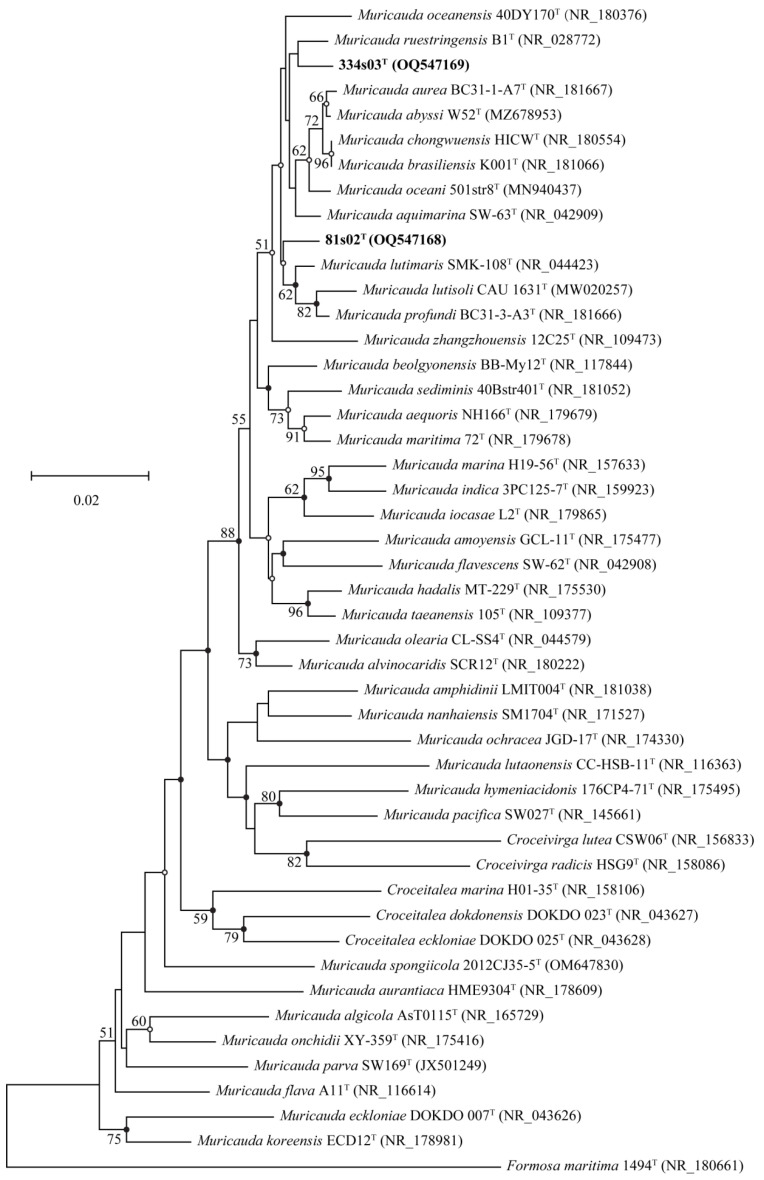
Phylogenetic tree, based on the 16S rRNA gene sequences using the neighbor-joining algorithm showing the position of strains 81s02^T^ and 334s03^T^. GenBank accession numbers used are given in the parentheses. Open circles indicated that the branches were recovered in maximum likelihood or maximum parsimony analyses. Filled circles showed branches that were recovered in maximum likelihood and maximum parsimony analyses. Bootstrap values higher than 50% were indicated at branch nodes. *Formosa maritima* 1494^T^ was used as an outgroup. Bar, 0.02 substitutions per nucleotide position.

**Figure 2 microorganisms-11-01580-f002:**
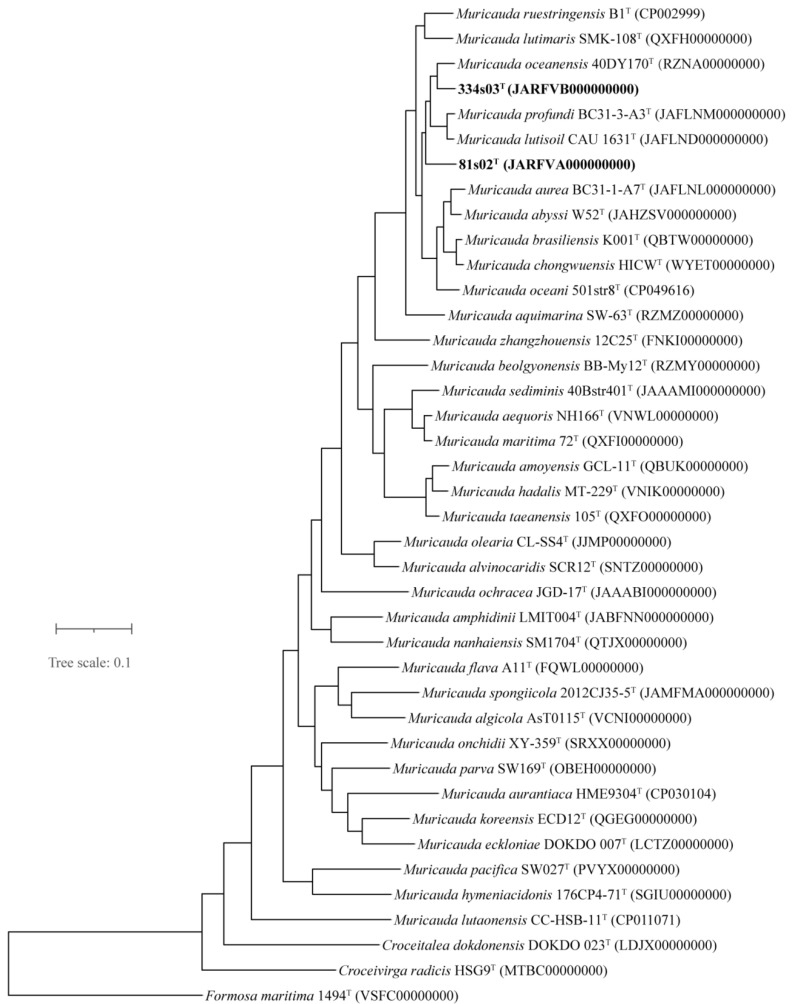
Phylogenomic tree of new strains (81s02^T^ and 334s03^T^) and other available type strains of the most closely related species based on 120 conserved concatenated proteins (Bac120 sets) from genomes sequences from the NCBI GenBank database. GenBank accession numbers used were given in the parentheses. *Formosa maritima* 1494^T^ was used as an outgroup. Bar, 0.1 substitutions per amino acid position.

**Figure 3 microorganisms-11-01580-f003:**
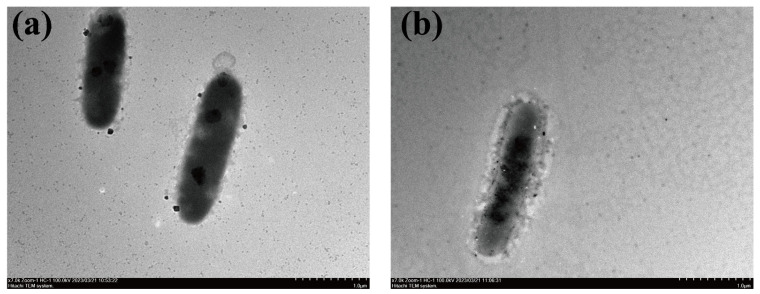
Cell micrograph of strains 81s02^T^ and 334s03^T^ grown in marine ZoBell broth for 3 days taken using transmission electron microscopy (Hitachi-HT7700). (**a**) 81s02^T^; (**b**) 334s03^T^.

**Figure 4 microorganisms-11-01580-f004:**
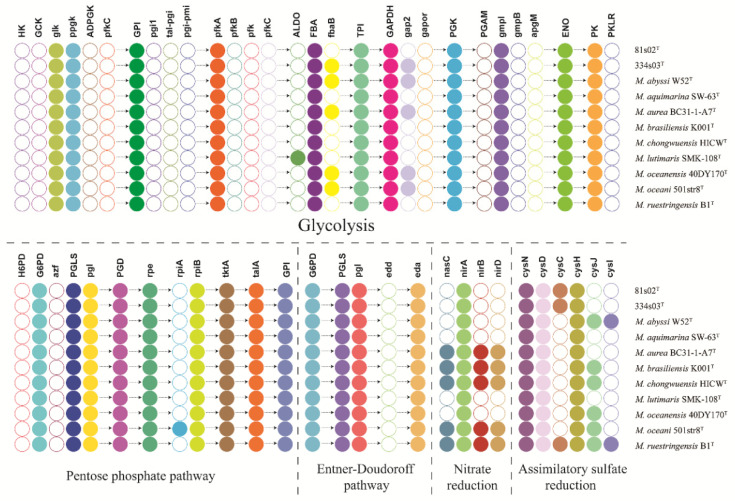
Presence and absence of metabolic pathway genes in eleven *Muricauda* strains. The hollow circles indicate no gene-related functions found in the genomes. The solid circles represent genes found in the genomes.

## Data Availability

The datasets used during the current study are available from the corresponding author on reasonable request.

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
