# Peer review of "Muricauda okinawensis sp. Nov. and Muricauda yonaguniensis sp. Nov., Two Marine Bacteria Isolated from the Sediment Core near Hydrothermal Fields of Southern Okinawa Trough"

_microorganisms, 2023, doi:10.3390/microorganisms11061580_

Round 1
Reviewer 1 Report
From line 142-145 explain cultivation methods in anaerobiosis condition??
What was used to reduce the medium and make it negative??Which indicator did you use??
In 3.2 paragraph there isn't resilts about growth in anaerobic and microaerophilic conditions. Explain better
TEM is the best way to highlight cellular ultrastructure....here images are not appreciable. In any case, i would recommend to put them in the main text.
i would explain better in the intro and discussion sections, about Muricauda in hydrothermal sites. In biblio there are evidence in hydrothermal sites also metal-tolerant and antibiotic resistent
Author Response
Point 1: From line 142-145 explain cultivation methods in anaerobiosis condition??
Response 1: Thanks for the question. In paragraph 2.4., authors described that “Anaerobic growth (10% CO2, 5% H2 and 85% N2) and micro-aerobic growth (5% O2, 10% CO2 and 85% N2) were detected in an anaerobic chamber in an atmosphere consisting of on MA with or without 0.25% (w/v) NaNO3 for 15 days at 30 ℃. ” This sentence is confusing. The words “an atmosphere consisting of” were deleted.
In detail for the method, strains inoculated on MA plates with or without 0.25% (w/v) NaNO3 and transfer to anaerobic culture tank. Then, the anaerobic and microaerobic were established by a gas control system (ANOXOMAT MARK II). Subsequently, the plates were incubated for at least two weeks.
Point 2: What was used to reduce the medium and make it negative??Which indicator did you use??
Response 2: Did the reviewer referring to the nitrate reduction? The medium used for this experiment is that “peptone 5.0g, beef extract 3g, KNO3 1g, Nacl 30g, water 1000 ml”. The indicators were sulfanilic acid-naphthylamine solution (A) and diphenylamine reagent (B). The suspension of the strains (3days and 7 days) were transferred on a colorimetric porcelain plate, then add solutions A and B successively. If there was no color change, it was considered negative for nitrate reduction.
Point 3: In 3.2 paragraph there isn't resilts about growth in anaerobic and microaerophilic conditions. Explain better.
Response 3: The result about growth in anaerobic and microaerophilic conditions has been added as a short brief description in paragraph 3.2.
Point 4: TEM is the best way to highlight cellular ultrastructure....here images are not appreciable. In any case, i would recommend to put them in the main text.
Response 4: The TEM pictures have been put in the main text as Figure 3.
Point 5: I would explain better in the intro and discussion sections, about Muricauda in hydrothermal sites. In biblio there are evidence in hydrothermal sites also metal-tolerant and antibiotic resistant
Reviewer 2 Report
The manuscript entitled “Muricauda okinawensis sp. Nov. and Muricauda yonaguniensis sp. Nov., Two Marine Bacteria Isolated from the Sediment Core Near Hydrothermal Fields of Southern Okinawa Trough” by Cao et al. is written clearly and organized. It presents an interesting study about the isolation and characterization of two novel strains from the southern Okinawa Trough. The descriptions of the strains, including their morphological, physiological, and genomic characteristics, are detailed and well-presented. The methods used for the phylogenomic analysis, such as the ANI, dDDH, and 16S rRNA gene sequence similarity, are appropriate and well-conducted.
However, one issue is that the species description is based only on a single strain for each species. It would be more comprehensive if additional strains of these species were also characterized, as this would strengthen the description and robustness of the new species. Overall, the work is of high quality and offers valuable additions to the microbial taxonomy field.
Author Response
Point 1: The manuscript entitled “Muricauda okinawensis sp. Nov. and Muricauda yonaguniensis sp. Nov., Two Marine Bacteria Isolated from the Sediment Core Near Hydrothermal Fields of Southern Okinawa Trough” by Cao et al. is written clearly and organized. It presents an interesting study about the isolation and characterization of two novel strains from the southern Okinawa Trough. The descriptions of the strains, including their morphological, physiological, and genomic characteristics, are detailed and well-presented. The methods used for the phylogenomic analysis, such as the ANI, dDDH, and 16S rRNA gene sequence similarity, are appropriate and well-conducted.
Response 1: Thank you for your recognition of our work
However, one issue is that the species description is based only on a single strain for each species. It would be more comprehensive if additional strains of these species were also characterized, as this would strengthen the description and robustness of the new species. Overall, the work is of high quality and offers valuable additions to the microbial taxonomy field.
Response 2: Thank you for your recommendation. As you said, it would be more comprehensive if additional strains of these species were also characterized. Unfortunately, only single strain for each species was isolated. In the future, we hope to isolate more microbial resources from marine environments.